# Alginate-Derived Elicitors Enhance β-Glucan Content and Antioxidant Activities in Culinary and Medicinal Mushroom, *Sparassis latifolia*

**DOI:** 10.3390/jof6020092

**Published:** 2020-06-25

**Authors:** Yong-Woon Kim, Yuanzheng Wu, Moon-Hee Choi, Hyun-Jae Shin, Jishun Li

**Affiliations:** 1Department of Biochemical and Polymer Engineering, Chosun University, Gwangju 61452, Korea; ywkim1205@naver.com (Y.-W.K.); aamoony1222@naver.com (M.-H.C.); 2Shandong Provincial Key Laboratory of Applied Microbiology, Ecology Institute, Qilu University of Technology (Shandong Academy of Sciences), Jinan 250103, China; wuyzh@sdas.org (Y.W.); yewu2@sdas.org (J.L.)

**Keywords:** Alginate, β-glucan, oligosaccharides, elicitation, *Sargassum* species, *Sparassis latifolia*, polyphenol, antioxidant

## Abstract

This study aimed to investigate the elicitation effects of alginate oligosaccharides extracted from brown algae (*Sargassum* species) on β-glucan production in cauliflower mushroom (*Sparassis latifolia*). Sodium alginate was refined from *Sargassum fulvellum*, *S. fusiforme*, and *S. horneri*, and characterized by proton nuclear magnetic resonance spectroscopy (^1^H NMR), resulting mannuronic acid to guluronic acid (M/G) rationes from 0.64 to 1.38. Three oligosaccharide fractions, ethanol fraction (EF), solid fraction (SF), and liquid fraction (LF), were prepared by acid hydrolysis and analyzed by Fourier transform infrared (FT-IR) spectra and high-performance anion-exchange chromatography with a pulsed amperometric detector (HPAEC-PAD). The samples of *S. fusiforme* resulted in the highest hydrolysate in SF and the lowest in LF, which was consistent with its highest M/G ratio. The SF of *S. fusiforme* and LF of *S. horneri* were chosen for elicitation on *S. latifolia*, yielding the highest β-glucan contents of 56.01 ± 3.45% and 59.74 ± 4.49% in the stalk, respectively. Total polyphenol content (TPC) and antioxidant activities (2,2’-Azino-bis(3-ethylbenzthiazoline-6-sulfonic acid) (ABTS) radical scavenging and Superoxide dismutase (SOD)-like activity) of aqueous extracts of *S. latifolia* were greatly stimulated by alginate elicitation. These results demonstrate that alginate oligosaccharides extracted from brown algae may be useful as an elicitor to enhance the nutritional value of mushrooms.

## 1. Introduction

Mushrooms have been recognized as medicine sources and functional foods since ancient times owing to their bioactive compounds and diverse health benefits [1]. Cauliflower mushrooms, species of *Sparassis* Fr., are culinary and medicinal mushrooms that primarily, but not exclusively, grow on the stumps of coniferous trees and are widely distributed throughout northern temperate forests [2]. The Asian *Sparassis* isolate was originally known as *S. crispa*, until morphological and molecular studies redefined it as *S. latifolia* [3,4]. Recently this mushroom has become cultivable in Japan and Korea using conifers [5]. The fruiting bodies of *S. crispa* and *S. latifolia* exhibit excellent effects for enhancing cytokine synthesis and preventing human diseases, such as gastric ulceration, oesophageal cancer, hypertension, and diabetes. Such effects are attributable to different compounds including polysaccharides, terpenoids, phenolic compounds, and glycoproteins [6,7,8].

As major constituents of fungal cell walls, β-glucans are present in all mushroom species and play important roles in their beneficial properties for human health [9]. Previous studies described a wide range of mushroom β-glucans with different structures, such as linear 1-3-β-glucans isolated from *Poria cocos*, linear 1-6-β-glucans from *Agaricus* spp., 6-branched 1,3-β-glucans from *Lentinus edodes* (designated lentinan) and *Grifola frondosa* (designated grifolan), and 3-branched 1,6-β-glucans from *Sarcodon aspratus* [10,11,12]. Among most studied fungal glucans, 6-branched 1,3-β-glucans were reported to be efficient as biological response modifiers (BRM) for the treatment of cancer and infectious diseases [13]. *Sparassis* species contain considerably higher contents of β-glucan than other mushrooms, up to 43.6% of dry weight in the fruiting bodies of *S. crispa* [14,15]. The primary structure of β-glucan isolated from *S. crispa* was 6-branched 1,3-β-glucan with one branch in every third main-chain unit, showing high water solubility and an estimated molecular weight of ca. 510 kDa [16,17]. Extraction and purification of β-glucans from mushroom mycelia and fruiting bodies have been established [18,19]. However, there are few studies about the enhancement of β-glucan contents in mushrooms.

To enhance the production of desirable compounds in plants or microorganisms, elicitation has been employed using biotic or abiotic elicitors to stimulate the accumulation of secondary metabolites [20,21]. For example, physical elicitation and enzyme treatments have been exploited for the growth enhancement and promotion of β-glucan contents in *S. latifolia* [22,23]. Alginate is a biotic elicitor that has been demonstrated to provide induced plant defense against pathogens, tolerance improvement to environmental stresses such as drought and salinity, and growth enhancement in *Eucomis autumnalis*, *Vitis vinifera*, *Phoenix dactylifera*, and *Olea europaea* [24,25,26,27]. Chemically, alginate is a linear copolymer consisting of homopolymeric blocks of residues of (1-4)-linked β-d-mannuronate (M) and its C-5 epimer α-l-guluronate (G), covalently linked together in different sequences [28]. As a biocompatible and immunogenic polysaccharide, alginate can be extracted from brown algae such as *Ecklonia* spp., *Laminaria* spp., or *Sargassum* spp. [29]

In this study, the physicochemical characteristics of alginate oligosaccharides extracted from *Sargassum* species from Korea and the enhancement of β-glucan content in *S. latifolia* by alginate elicitation were investigated. To the best of our knowledge, there has been no prior attempt to investigate alginate-derived elicitors for enhancement of β-glucan synthesis in mushrooms.

## 2. Materials and Methods

### 2.1. Collection of Marine Brown Algae

Three brown algae of *Sargassum* species were collected from the Korean coast. *S. fulvellum* (Turner) C. Agardh was collected in March 2017 at Jindo (34°22′18.0″ N, 126°08′09.4″ E), *S. fusiforme* (Harvey) Setchell was harvested in July 2017 at Wando (34°20′11.8″ N, 126°48′54.6″ E), and *S. horneri* (Turner) C. Agardh was collected in February 2018 at Jeju (33°31′08.3″ N, 126°31′16.7″ E). All *Sargassum* samples were cleaned by washed in distilled water to remove sand and excess salt. Samples were then oven dried at 60 °C. The samples were stored in plastic bags at room temperature (25 °C) until alginate extraction.

### 2.2. Extraction of Sodium Alginate

Sodium alginate was extracted as described by Davis et al. [30] In brief, 25 g of *Sargassum* samples were soaked in 800 mL of 2% formaldehyde for 24 h at room temperature, washed with distilled water, added to 800 mL of 0.2 M HCl, and left for 24 h. The samples were then washed with distilled water before extraction with 2% sodium carbonate (Na_2_CO_3_) for 3 h at 100 °C. The filtered extract was centrifuged (10,000 rpm, 30 min, 4 °C) and the supernatant was precipitated with 3 volumes of 95% ethanol. The precipitate was washed with acetone and dried at 60 °C.

### 2.3. Preparation of Oligosaccharide Fractions

Oligosaccharide separation from sodium alginate was performed as described by Asilonu et al. [31] This method gave three oligosaccharide fractions instead of two fractions from sodium alginate. In brief, 10 g of sodium alginate samples were acid hydrolyzed with 1 L of 0.3 M HCl in an autoclave at 121 °C for 1 h. The suspension was quickly cooled using ice water and neutralized with NaOH. Sodium chloride (NaCl) was added to a final concentration of 0.5% (*w/v*), and the solution mixture precipitated with 2 volumes of 95% ethanol and centrifuged (4000 rpm, 30 min, 4 °C). The supernatant was marked as ethanol fraction (EF). The precipitate was then washed with distilled water, centrifuged again, dissolved in distilled water, and adjusted to pH 2.85 with 1 M HCl. The solid fraction (SF) and liquid fraction (LF) were separated and lyophilized. Thus, three different oligosaccharide fractions (EF, SF, and LF) were obtained for further analysis.

### 2.4. Physicochemical Analysis of Oligosaccharide Fractions

#### 2.4.1. Molecular Weight Analysis

The average molecular weight (*M*_W_) of the EF, LF, and SF oligosaccharide fractions was determined using an SB-803 HQ column (6 µm, 8.0 mm × 300 mm, Shodex, Japan) and an SB-805 HQ column (13 µm, 8.0 mm × 300 mm, Shodex, Japan). A mobile phase of 0.1 mol/L NaCl solution was used after 0.22 µm filtration. The flow rate was 0.5 mL/min and analyses were performed at 50 °C. The sodium alginate sample (100 mg) was then resuspended in 10 mL distilled water. Molecular weights were determined by reference to a calibration curve using pullulan standards [32].

#### 2.4.2. H NMR Spectroscopy Analysis

Proton nuclear magnetic resonance (^1^H NMR) spectroscopy was acquired on alginate solution (1% *w/v*) in D_2_O, with recordings at 80 °C using an AVANCE III HD 400 (Bruker Scientific Instruments, USA) spectrometer operating at a frequency of 400 MHz. The individual blocks of guluronic and mannuronic acids (*F_G_* and *F_M_*, respectively), the homopolymeric (*F_GG_* and *F_MM_*, respectively) blocks, and the heterogeneous (*F_GM_* or *F_MG_*) blocks of alginate were calculated using the areas of I, II, and III (*A_I_*, *A_II_*, and *A_III_*, respectively) signals according to the Grasdalen [33] method and Equations (1)–(3):(1)$$F_{G}=\frac{A_{I}}{A_{II}+A_{III}}$$
(2)$$F_{M}=1-F_{G}$$
(3)$$M/G=\frac{1-F_{G}}{F_{G}}=\frac{F_{M}}{F_{G}}$$

The double fractions (*F_GM_* and *F_MM_*) were deduced by referring to Equations (4)–(6):(4)$$F_{GG}=\frac{A_{II}}{A_{II}+A_{III}}$$
(5)$$F_{GM}=F_{MG}=F_{G}-F_{GG}$$
(6)$$F_{MM}=F_{M}-F_{MG}$$

#### 2.4.3. FT-IR Spectroscopy Analysis

Fourier transform infrared (FT-IR) spectra were measured on a Nicolet 6700 FT-IR (Thermo Electron Co., Waltham, MA, USA). Spectra of the alginate samples in KBr pellets were recorded in the 4000–400 cm^−1^ range.

#### 2.4.4. Monosaccharide Analysis

Monosaccharide analysis for the three oligosaccharide fractions was performed as follows [34]. The first acid hydrolysis was performed in 3 mL of 72% H_2_SO_4_ with 0.3 g of sample for 2 h at 30 °C. The second hydrolysis was carried out by adding 84 mL of distilled water to the first hydrolysate followed by acid hydrolysis at 121 °C for 1 h in an autoclave. Analysis was performed using high performance anion exchange chromatography with a pulsed amperometric detector (HPAEC-PAD, ICS-5000, Dionex Co., Sunnyvale, CA, USA) equipped with a current detector, with CarboPac PA-4 (250 mm × 4 mm, Dionex Co., USA) used as the column at room temperature. An 18 mM NaOH solution was chosen as the mobile phase with a flowrate of 1.0 mL/min.

### 2.5. Cultivation of Sparassis Latifolia

*Sparassis latifolia* JF02-06 was provided by Jeonnam Forest Research Institute (Naju, Korea). A sawdust-based medium was used for the cultivation of cauliflower mushrooms as adopted by Park et al. [23] Fermented sawdust of *Larix kaempferi*, corn, and wheat flour were mixed at a ratio of 8:1:1 (*w/w/w*), followed by the addition of 10% aqueous solution of starch syrup and an adjustment of the moisture content to 55–60%. The medium was packed in a 500 mL incubation bottle and sterilized at 121 °C for 90 min. The inoculum of *S. latifolia* was prepared in potato dextrose broth (PDB) for 3 weeks and 10 mL of liquid inoculum was inoculated into each bottle. The inoculated medium was then incubated at room temperature. Cultivation was completed after 50–70 days from inoculation. When mycelium was formed in the medium, the culture was transferred to the cultivation room at a temperature of 20 ± 2 °C and 95% humidity. Fruit-shaped fruiting bodies were formed after 1 week and then harvested after 40–45 days.

### 2.6. Elicitation by Alginate Oligosaccharide Fractions

Two of the oligosaccharide fractions, LF and SF (each 200 mg/L in distilled water), were applied to the surface of the sawdust medium for elicitation: 12 mL/bottle after sterilization of the medium (first application) and 20 mL/bottle after transfer to the cultivation room (second application). The same volume of distilled water was used as a control group. All treatments were arranged in a completely randomized block design with four replicates. Fruiting body and stalk of *S. latifolia* were harvested, frozen in liquid nitrogen, and stored in a freezer at −80 °C until extraction and analysis.

### 2.7. Assay of Glucan Content

Content of total and β-glucans was determined by a β-glucan assay kit (Cat. No. K-YBGL, Megazyme International, Wicklow, Ireland), following the H_2_SO_4_ acid hydrolysis procedure by McCleary and Draga [35]. Briefly, after being milled to pass through a 1.0 mm screen, 100 mg of the dried mushroom samples were added with 2 mL of ice-cold 12 M H_2_SO_4_, and then the mixture was stirred vigorously and incubated in an ice-cold water bath for 2 h. Then, 12 mL of distilled water was added to each sample and the suspension was kept in a boiling-water bath (~100 °C) for 2 h. After cooling to room temperature, 6 mL of 10 M KOH was added and the volume was adjusted to 100 mL with 200 mM sodium acetate buffer (pH 5.0). After centrifugation at 12,000 rpm for 10 min, an aliquot of the supernatant (0.1 mL) was mixed with 0.05 mL of exo-1,3-β-glucanase (20 U/mL) plus β-glucosidase (4 U/mL) and incubated at 40 °C for 60 min. Then, the mixture was incubated at 40 °C for 20 min with 3 mL of glucose-oxidase/peroxidase-reagent (GOPOD). Total glucan content was evaluated by a UV-Vis spectrophotometer (S-3100, SCINCO, Seoul, Korea) at λ = 510 nm with the reagent blank.

The α-glucan content was determined after incubation at 40 °C for 30 min of the suspension of 100 mg mushroom samples in 2 mL of 2 M KOH with an addition of 8 mL of 1.2 M sodium acetate buffer (pH 3.8) with 0.2 mL of amyloglucosidase (1630 U/mL) plus invertase (500 U/mL). Each sample was centrifuged at 12,000 rpm for 10 min, and 0.1 mL of the supernatant was analyzed for glucose by mixing with 0.1 mL of 200 mM of sodium acetate buffer (pH 5.0) and 3 mL of GOPOD. α-Glucan content was evaluated by a UV-Vis spectrophotometer. β-Glucan content was determined as the difference between total and α-glucan contents.

### 2.8. Determination of Total Polyphenol Content

Total polyphenol content (TPC) of aqueous extracts of *S. latifolia* after alginate elicitation was determined by the modified Folin-Ciocalteu method [36]. An amount of 0.5 mL of the extract was mixed with 0.5 mL of 0.2 N Folin-Ciocalteu reagent and 0.5 mL of 2% (*w/v*) sodium carbonate. The mixture was vortexed and incubated at room temperature for 30 min. The absorbance of the mixture was measured at λ = 750 nm with a UV-Vis spectrophotometer. Total polyphenol content was expressed as of gallic acid equivalent (GAE) mg/100 g.

### 2.9. Measurement of Antioxidant Activities

The antioxidant activities of aqueous extracts of *S. latifolia* were evaluated by scavenging activity of 2,2′-azino-bis-3-ethylbenzothiazoline-6-sulphonic acid (ABTS) radical cation and superoxide dismutase (SOD)-like activity.

#### 2.9.1. Assay of ABTS Radical Scavenging Activity

2,2’-Azino-bis(3-ethylbenzthiazoline-6-sulfonic acid) (ABTS) radical scavenging activity was measured using a modified method of Re et al. [37] The ABTS stock solution was prepared by mixing 7 mM ABTS with 2.45 mM potassium persulfate and then kept in the dark for 12 h. The stock solution was then diluted with phosphate buffered saline (PBS, pH 7.4) until absorbance reached 0.80 ± 0.02 at λ = 730 nm using a UV-Vis spectrophotometer. Of the extracts of *S. latifolia*, 0.2 mL was mixed with 1 mL of diluted ABTS solution and left for 15 min in the dark. ABTS radical scavenging activity (%) was calculated as in Equation (7) as follows:(7)$$\mathrm{ABTS}\;\mathrm{radical}\;\mathrm{scavenging}\;\mathrm{activity}\;\left(\%\right)=\frac{A_{0}-A_{1}}{A_{0}}\times 100,$$
where *A*_0_ is the absorbance of the blank sample using water, and *A*_1_ is the absorbance of the aqueous extracts of *S. latifolia*.

#### 2.9.2. Assay of SOD-Like Activity

Superoxide dismutase (SOD)-like activity was measured by reference using an SOD kit (Cat. No. 19160, Sigma-Aldrich, St. Louis, MO, USA) [38]. Of the aqueous extracts of *S. latifolia*, 20 µL was mixed with 200 µL of water-soluble tetrazolium salt (WST) working solution and 20 µL of enzyme working solution to each well in a 96-well microplate. Then the microplate was incubated at 37 °C for 20 min. The absorbance of each sample was measured at λ = 450 nm using a microplate reader. The SOD-like activity (%) was calculated as in Equation (8) as follows:(8)$$\mathrm{SOD}-\mathrm{like}\;\mathrm{activity}\;\left(\%\right)=\frac{\left(\mathrm{Blank}\;1-\mathrm{Blank}\;3\right)-\left(\mathrm{Sample}-\mathrm{Blank}\;2\right)}{\left(\mathrm{Blank}\;1-\mathrm{Blank}\;3\right)}\times 100$$
where Blank 1 is the absorbance of the water with WST working solution and enzyme working solution, Blank 2 is the absorbance of sample with WST working solution and dilution buffer, Blank 3 is the absorbance of water with WST working solution and dilution buffer, and Sample is the absorbance of the aqueous extracts of *S. latifolia*.

### 2.10. Statistical Analysis

Data were analyzed using analysis of variance (ANOVA) followed by Duncan’s multiple-range test (*p* < 0.05) using SPSS software (version 23, SPSS, Chicago, IL, USA). Error bars indicate the mean ± SD and different letters describe significant differences within the same application data group.

## 3. Results

### 3.1. Physicochemical Properties of Alginate

Sodium alginate was extracted and refined from three *Sargassum* species, namely *S. fulvellum*, *S. fusiforme*, and *S. horneri*. The composition of extracted alginate is summarized in Table 1. The highest alginate yield was obtained from *S. fusiforme* (37.84 ± 0.48%), followed by 34.11 ± 1.65% from *S. horneri* and 30.88 ± 1.51% from *S. fulvellum*. The size of the extracted alginates was relatively high in *S. fusiforme* and *S. fulvellum* with an average *M*_W_ of 504.65 and 461.07 kDa, respectively. However, the alginate from *S. horneri* had a much lower *M*_W_ (138.10 kDa) than that of the other two algae. This indicates that different ranges of alginates are obtained from different *Sargassum* species.

The ratio of mannuronic acid to guluronic acid (M/G) of the structural blocks were determined by ^1^H NMR analysis. As shown in Table 1, the M/G ratio of *Sargassum* species ranged from 0.64 to 1.38, with the highest ratio of 1.38 from *S. fusiforme* and similar M/G rationes between *S. fusiforme* and *S. horneri*. The guluronic acid anomeric proton (G-1) occurred at 5.56–5.58 ppm (peak Ι), guluronic acid H-5 (G-5) occurred at 4.96–4.98 ppm (peak ΙΙΙ), and mannuronic acid anomeric proton (M-1) and the C-5 of alternating blocks (GM-5) overlapped at 5.21–5.23 ppm (peak ΙΙ), as shown in Figure 1.

The FT-IR spectrum of sodium alginate of *Sargassum* species is presented in Figure 2. A broad band at 3466.25 cm^−1^ was assigned to the hydrogen-bonded (O-H) stretching vibrations, and a weak signal at 2928.43 cm^−1^ was attributed to C-H stretching vibrations. The peaks at 1619.79 cm^−1^ and 1427.52 cm^−1^ were attributed to asymmetric stretching of carboxylate O-C-O vibrations. The characteristic peaks of alginate include 1096.76 cm^−1^ assigned to β-mannuronic acid and 1029.66 cm^−1^ assigned to α-L-guluronic acid. Moreover, signals of glucuronic acid (1737.01, 1629.72, and 1144.66-937.31 cm^−1^) could be detected in Figure 2 (LF and EF of a,b,c).

### 3.2. Oligosaccharide Analysis

Sodium alginates extracted from *Sargassum* species were submitted to a two-step H_2_SO_4_ acid hydrolysis process to obtain partial hydrolysates for monosaccharide analysis. The block distribution of partial acid hydrolysates in EF, SF, and LF were 13.9–20.4%, 15.8–22.8%, and 61.1–65.0%, respectively. The block distribution of this study was compared with that of previous reports by Haug et al. and Leal et al. [39,40], as summarized in Table 2. The comparison data showed a great difference in partial hydrolysis yield according to species used for the preparation of alginate oligosaccharides. The samples of *S. fusiforme* showed the highest hydrolysate yields in SF and the lowest in LF among *Sargassum* species, which was consistent with the highest M/G ratio of 1.38 and might be explained by the high content of polymannuronic acid and low content of polyguluronic acid. *S. fulvellum* and *S. horneri* both produced lower hydrolysate yields in SF and higher yields in LF, as indicated by their low M/G rationes. All three *Sargassum* species yielded similar percentages of hydrolysates in EF.

HPAEC-PAD analysis of the H_2_SO_4_ hydrolysate of sodium alginate showed that monosaccharides were dominated by the weight of fucose, rhamnose, arabinose, galactose, glucose, mannose, xylose, mannuronic acid, guluronic acid, and glucuronic acid (Table 3). The major alginate contents comprised mannuronic acid and guluronic acid, as expected. The content of uronic acids was almost the same as that of *S. turbinarioides* as reported by Fenoradosoa et al. [41] The main monosaccharides of alginate from EF, SF, and LF were mannuronic acid, guluronic acid, and glucuronic acid. The size of the extracted ethanol fraction ranged from 650 to 5500 Da.

### 3.3. Elicitation by Alginate on β-Glucan Contents

From the results of oligosaccharide analysis of different fractions of *Sargassum* species, the solid fraction (SF) of *S. fusiforme* and liquid fraction (LF) of *S. horneri* were chosen as elicitors because of their high content of mannuronic acid and guluronic acid, respectively. The elicitation on *Sparassis latifolia* showed that SF treatment required a cultivation period of 47 days for mycelial growth, whereas LF treatment and the control group required a cultivation period of 43 days (Figure 3a). SF treatment provided higher mushroom production (197.30 ± 2.64 g) than the control group (192.04 ± 1.58 g), whereas LF treatment (191.05 ± 2.43 g) showed slightly lower results than the control group (Figure 3b). However, an additional spray of SF and LF elicitors resulted in lower yields of mushroom production, and this may be caused by excessive humidity as the control group also showed a decline with the second application.

These alginate-derived elicitors presented different effects on β-glucan content in the fruiting body and stalk of *S. latifolia*. SF elicitor was more effective for β-glucan enhancement in the fruiting body, whereas LF elicitor was more effective in the stalk (Figure 3c). For the second application, both SF (56.01 ± 3.45%) and LF (59.74 ± 4.49%) treatments showed obvious high increments of β-glucan contents in the stalk compared with the control (21.65%), which indicates that the additional spray of alginate oligosaccharide fractions apparently stimulated the β-glucan synthesis process in the stalk compared with that in the fruiting body (Figure 3d).

### 3.4. Alginate Effects on Total Polyphenol Content

After the second application of alginate oligosaccharide fractions, total polyphenol content (TPC) in the fruit body and in the stalk of *S. latifolia* was determined, and gallic acid was used as standard. The TPC of the samples ranged from 140.11 ± 6.72 to 420.87 ± 16.99 GAE mg/100g (Figure 4). There were significant differences in TPC (*p* < 0.05) depending on the mushroom part and treated elicitor. The TPC of stalk samples was higher than for the corresponding fruit body samples. LF treatment exhibited a higher TPC value than SF treatment, both in the fruit body and stalk samples.

### 3.5. Alginate Effects on Antioxidant Activities

The antioxidant activities of aqueous extracts of *S. latifolia* after alginate elicitation were measured by ABTS radical scavenging and SOD-like activity. ABTS radical scavenging ability was calculated by IC_50_ value. The results ranged from 926.67 ± 56.26 to 4002.19 ± 149.53 µg·mL^−1^ (Figure 5a). SOD-like activity was measured by a colorimetric method and calculated by IC_50_ value. IC_50_ values of SOD-like activity ranged from 6.74 ± 0.29 to 18.70 ± 0.57 mg·mL^−1^ (Figure 5b). The highest ABTS radical scavenging and SOD-like activity were recorded with LF elicitor treated stalk and the lowest were from the fruit body of the control group. Similar to the TPC results, both fruit body and stalk samples of LF treatment exhibited higher antioxidant activities than the corresponding samples treated by SF elicitor. The results of antioxidant activities were significantly (*p* < 0.05) stimulated with the alginate elicitation.

## 4. Discussion

*Sparassis* species are widely used medicinal mushrooms in traditional Chinese medicine. However, despite its commercial potential, cultivation of *S. latifolia* has been limited to a few high-tech commercial farms because of its slow mycelial propagation into solid medium [42]. In addition, its slow rate of growth is a major obstacle to the employment of *S. latifolia* as a producer of β-glucan. Biotic or abiotic elicitors have been employed to enhance the production of β-glucan in *S. latifolia*. Ryoo et al. exploited the effects of physical stimulation of UV irradiation and temperature shock on β-glucan contents in *S. latifolia* and found that β-glucan yields reached 41.36 ± 2.96% of flabella and 42.16 ± 2.90% of stipe after UV irradiation for 10 min [22]. Park et al. utilized chitinase, β-glucuronidase, and lysing enzyme complex as elicitors to enhance the β-glucan content of *S. latifolia* and produced an increase in β-glucan concentration of 31%, although the treatment caused a decrease in mushroom yield [23]. These results indicate that the elicitation technique needs to be evaluated for application in high-value mushroom cultivation.

This study aimed to increase the contents of β-glucan in *S. latifolia* through alginate elicitation, which was extracted from *Sargassum* species. Marine algae are an excellent biomass source used as a fertilizer for soil reformation because they are rich in carbohydrates and mineral content [43]. However, although *Sargassum* species contain a large amount of alginate and fucoidan, the high content of arsenic makes them unsuitable for consumption [44,45]. Polysaccharide-derived hydrocolloids in marine algae such as alginate, carrageenan, fucoidan, laminarin, ulvan, and glucuronan have been suggested as biotic elicitors to induce several different mechanisms including host defense mechanisms and growth enhancement in plant and fungi [46,47,48,49].

Alginate is the major structural polysaccharide of *Sargassum* species, consisting of MM, GG, and MG blocks arranged in various proportions in block units. The alginate M/G ratio and yield may vary in accordance with the algae harvest season and geographical location [41]. Leal et al. studied block fractions of alginate in the brown algae *Lessonia flavicans* and *Desmarestia ligulata* [40]. The ethanol fraction obtained by partial acid hydrolysis of alginate was mainly composed of a heteropolymer block (MG). In addition, the solid fraction was rich in polymannuronic acid (MM) and the liquid fraction was rich in polyguluronic acid (GG) [28]. These were consistent with the results found in this study (see Table 2).

Alginate elicitors have been employed to modify cell metabolism in order to enhance the productivity of useful metabolites in plants and microorganisms [50]. The addition of chitosan, chitosan oligosaccharide, and alginate oligosaccharide to a culture of *Panax ginseng* C. A. Mey hairy roots caused growth inhibition and rises in total ginseng saponin accumulation with elicitor concentration [51]. Mannuronic acid of alginate may have a potent stimulatory effect on cytokine production, and it appeared to affect β-glucan content because of increased specific immunity owing to immunological effects [52]. In contrast, guluronic acid suppresses tissue damage caused by cytokines produced in response to inflammatory stimuli [53]. Our group investigated the supplement of sodium alginate to *S. latifolia* for the induced expression of γ-aminobutyric acid (GABA) both in the mycelia and fruiting bodies, which inhibited the dendrite outgrowth of excitatory neurons, but not that of inhibitory neurons [54]. Genome sequencing and genomics studies are underway to determine the underlying effect of alginate elicitors on β-glucan content of *S. latifolia* [55].

The effects of alginate elicitors on antioxidant activity and polyphenol synthesis in plants and microorganisms were also verified. Chitosan, pectin, and alginate promoted accumulation of phenolic acids, particularly 3-*O*-glucosyl-resveratrol, in *Vitis vinifera*, which was positively correlated with increased accumulation of anthocyanin [25]. Ulvan, carrageenan, alginate, and laminarin were examined for any elicitation effect in twigs of olive trees to elicit phenolic metabolism and control against *Verticillium* wilt of olive caused by *V. dahlia*, and the results showed increased phenylalanine ammonia-lyase (PAL) activity and total polyphenol content combined with the decline of wilt symptoms [27]. Sodium alginate extracted from Moroccan brown algae *Fucus spiralis* and *Bifurcaria bifurcata* was evaluated for elicitation on phenolic metabolism including PAL activity and total polyphenol content in seedling roots of date palm, and the results showed that PAL activity and phenolic compound content were stimulated with 1 mg·mL^−1^ sodium alginate [26].

The present study suggested that treatment with *Sargassum* alginate exhibited elicitation effects on the growth and β-glucan contents in *S. latifolia*. Elicitors derived from algae alginate can be widely used in the cultivation process of other culinary and medicinal mushrooms.

## 5. Conclusions

Based on our results, the products of partial acid hydrolysis of alginate oligosaccharides could act as elicitors for stimulated growth of *S. latifolia* and production of useful metabolites such as β-glucan and polyphenols.

## Figures and Tables

**Figure 1 jof-06-00092-f001:**
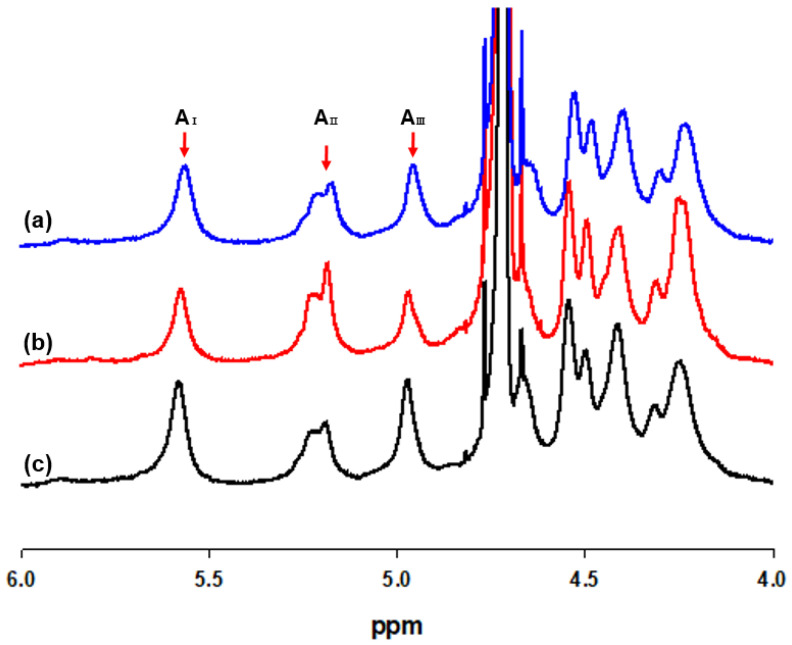
^1^H NMR spectra of sodium alginates extracted from (a) *Sargassum fulvellum*, (b) *S. fusiforme,* and (c) *S. horneri* (A_I_: Area of guluronic acid anomeric proton peak I; A_II_: Area of mannuronic acid anomeric proton and the C-5 of alternating blocks peak II; A_III_: Area of guluronic acid peak III).

**Figure 2 jof-06-00092-f002:**
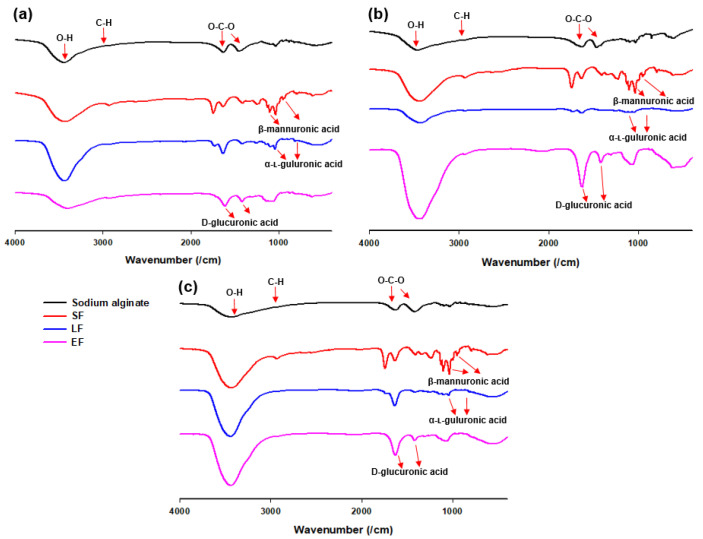
Fourier transform infrared (FT-IR) spectra of sodium alginate, solid fraction (SF), liquid fraction (LF), and ethanol fraction (EF) from (**a**) *Sargassum fulvellum*, (**b**) *S. fusiforme,* and (**c**) *S. horneri.*

**Figure 3 jof-06-00092-f003:**
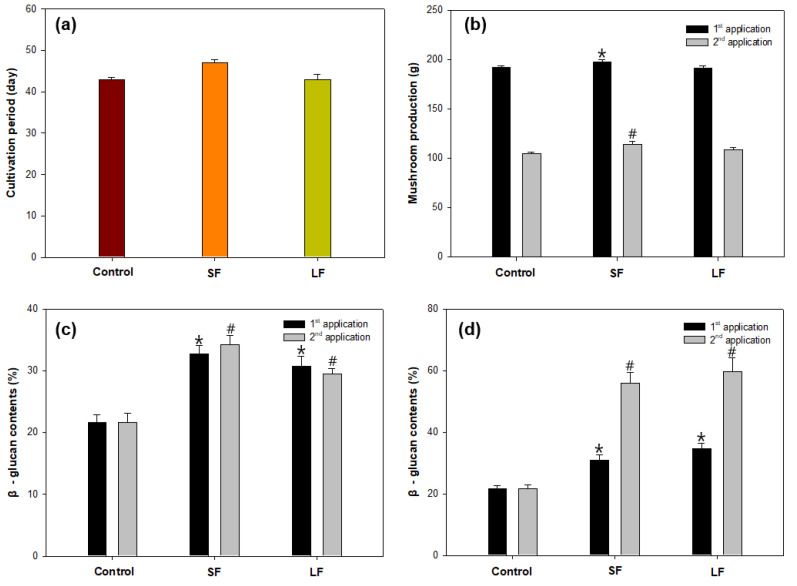
Elicitation of *Sparassis latifolia* by alginate oligosaccharide fractions on (**a**) cultivation period, (**b**) mushroom production, (**c**) β-glucan content of the fruiting body, and (**d**) β-glucan content of the stalk. SF: Solid fraction of *Sargassum fusiforme*; LF: Liquid fraction of *S. horneri*; Control: distilled water. (All data are presented as the mean ± SD. A *t*-test was used to compare the control and treated samples. * *p* < 0.05, vs. the control in first application; # *p* < 0.05, vs. the control in second application).

**Figure 4 jof-06-00092-f004:**
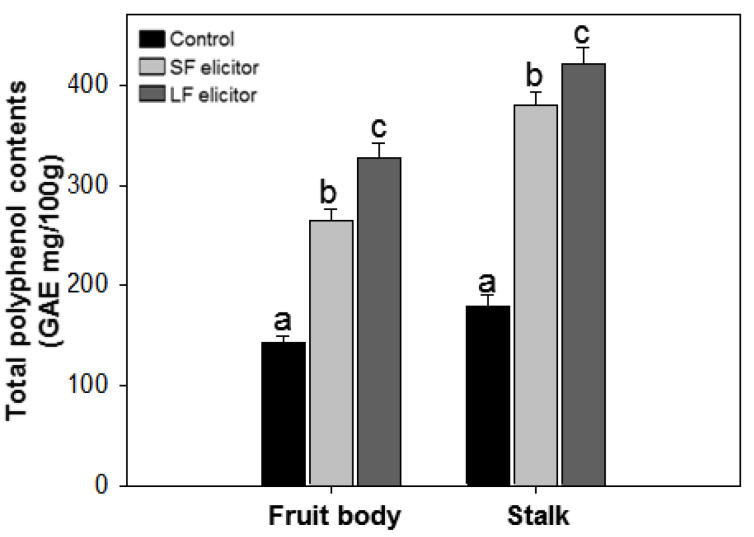
Total polyphenol contents of aqueous extracts of *Sparassis latifolia* after alginate elicitation. SF: Solid fraction of *Sargassum fusiforme*; LF: Liquid fraction of *S. horneri*; Control: distilled water. The data were analyzed using analysis of variance (ANOVA) followed by Duncan’s multiple-range test (*p* < 0.05) using SPSS software (SPSS, Chicago, IL). Error bars indicate the mean ± SD and different letters describe greater differences within the same part of *S. latifolia* (fruit body and stalk).

**Figure 5 jof-06-00092-f005:**
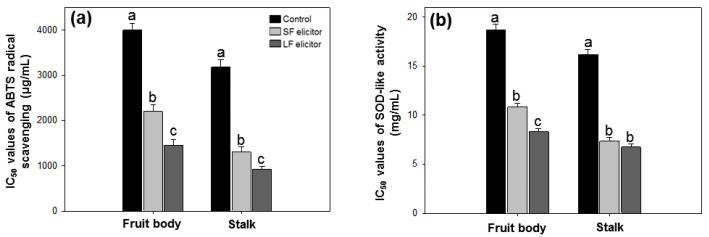
IC_50_ values of ABTS radical scavenging (**a**) and SOD-like activity (**b**) of aqueous extracts of *Sparassis latifolia* after alginate elicitation. SF: Solid fraction of *Sargassum fusiforme*; LF: Liquid fraction of *S. horneri*; Control: distilled water. The data were analyzed using analysis of variance (ANOVA) followed by Duncan’s multiple-range test (*p* < 0.05) using SPSS software (SPSS, Chicago, IL, USA). Error bars indicate the mean ± SD and different letters describe great differences within the same part of *S. latifolia* (fruit body and stalk).

**Table 1 jof-06-00092-t001:** Composition of sodium alginate extracted from *Sargassum* species.

Species	Yield (%)	*M*w (kDa)	M/G	*F_M_*	*F_G_*	*F_MM_*	*F_MG_*	*F_GG_*
*Sargassum fusiforme*	37.84 ± 0.48	504.65	1.38	0.58	0.42	0.55	0.03	0.39
*S. fulvellum*	30.88 ± 1.51	461.07	0.88	0.47	0.53	0.43	0.04	0.49
*S. horneri*	34.11 ± 1.65	138.10	0.64	0.39	0.61	0.30	0.09	0.52

*M*_W_: average molecular weight; *F_M_*: Mannuronic acid block fractions; *F_G_*: Guluronic acid block fractions; *F_MM_*: Homopolymeric mannuronic acid block fractions; *F_MG_*: Heterogeneous block fractions (mannuronic acid and guluronic acid); *F_GG_*: Homopolymeric guluronic acid block fractions.

**Table 2 jof-06-00092-t002:** Block distribution of sodium alginate from different species of brown algae.

Species	M/G	Partial Hydrolysis (Yields)			Reference
		EF (%)	SF (%)	LF (%)	
*Sargassum fusiforme*	1.38	63.8	20.4	15.8	This study
*S. fulvellum*	0.88	65.0	13.9	21.1	This study
*S. horneri*	0.64	61.1	16.1	22.8	This study
*Ascophyllum nodosum*	1.85	52	35	13	[39]
*Chordaria flagelliformis*	0.90	21	28	51	[39]
*Desmarestia aculeata*	0.85	27	23	50	[39]
*Dictyosiphon foeniculaceus*	0.85	25	25	50	[39]
*Fucus serratus*	1.30	35	34	31	[39]
*Laminaria digitata*	1.45	34	43	23	[39]
*L. hyperborea*, fronds	1.35	26	43	31	[39]
*L. hyperborea*, stripes	0.65	25	15	60	[39]
*Pelvetia canaliculata*	1.50	38	37	25	[38]
*Pylaiella*	0.75	40	18	42	[39]
*Scytosiphon lomentaria*	1.15	25	35	40	[39]
*Spermatochnus paradoxus*	1.30	35	32	33	[39]
*Desmarestia ligulata*	0.58	3.7	25.1	56.4	[40]
*D. ligulata*	0.77	3.5	37.0	47.1	[40]
*Lessonia flavicans*	1.03	8.5	41.3	22.2	[40]

EF: Ethanol fraction of acid-hydrolyzed sodium alginate; SF: Solid fraction at pH 2.85 (fraction enriched in polymannuronic acid); LF: Liquid fraction at pH 2.85 (fraction enriched in polyguluronic acid).

**Table 3 jof-06-00092-t003:** Monomeric carbohydrate contents from *Sargassum* species determined by high-performance anion-exchange chromatography with a pulsed amperometric detector (HPAEC-PAD) analysis after two-step sulfuric acid hydrolysis (mg·g^−1^).

Species	Fraction	Fuc	Rham	Arab	Gal	Glu	Man	Xyl	Mannu	Gulur	Glucu
*Sargassum fusiforme*	SA	26.38 ± 0.88	ND	ND	6.21 ± 0.16	0.29 ± 0.02	7.17 ± 0.25	4.50 ± 0.18	179.64 ± 2.37	167.57 ± 1.88	10.67 ± 0.29
	SF	0.32 ± 0.01	ND	ND	0.10 ± 0.01	ND	ND	ND	54.29 ± 0.14	13.18 ± 0.51	ND
	LF	1.84 ± 0.03	ND	ND	0.69 ± 0.01	ND	0.082 ± 0.04	0.35	8.47 ± 0.09	107.57 ± 1.88	ND
	EF	7.68 ± 0.10	ND	ND	2.8 ± 0.04	ND	3.59 ± 0.04	0.87 ± 0.02	5.46 ± 0.02	11.38 ± 0.11	6.39 ± 0.06
*S. fulvellum*	SA	24.21 ± 0.60	0.12 ± 0.01	0.03 ± 0.00	15.07 ± 0.32	0.35 ± 0.00	8.59 ± 0.36	4.60 ± 0.05	136.79 ± 1.82	167.19 ± 1.10	20.27 ± 0.22
	SF	0.17 ± 0.00	ND	ND	ND	0.11 ± 0.00	0.48 ± 0.00	0.05 ± 0.00	42.07 ± 2.82	7.53 ± 6.79	ND
	LF	3.88 ± 0.05	0.03 ± 0.00	0.05 ± 0.00	2.40 ± 0.03	1.77 ± 0.02	2.09 ± 0.04	1.08 ± 0.03	26.18 ± 0.77	100.16 ± 0.54	4.11 ± 0.31
	EF	8.86 ± 0.06	0.05 ± 0.00	0.01 ± 0.00	4.96 ± 0.04	0.10 ± 0.03	4.42 ± 0.04	1.03 ± 0.04	9.11 ± 0.18	3.56 ± 0.20	4.24 ± 0.05
*S. horneri*	SA	20.14 ± 0.20	ND	ND	13.45 ± 0.09	ND	5.70 ± 0.13	4.40 ± 0.24	127.37 ± 0.39	189.99 ± 0.36	16.93 ± 0.32
	SF	0.24 ± 0.00	ND	ND	ND	0.19 ± 0.00	0.14 ± 0.00	0.09 ± 0.00	43.98 ± 5.70	7.89 ± 1.05	ND
	LF	3.26 ± 0.02	0.03 ± 0.00	0.01 ± 0.00	2.51 ± 0.02	0.16 ± 0.02	2.11 ± 0.05	0.34 ± 0.02	22.21 ± 1.08	135.61 ± 3.18	3.44 ± 0.31
	EF	9.26 ± 0.12	0.08 ± 0.00	ND	6.57 ± 0.04	0.16 ± 0.00	5.52 ± 0.08	1.03 ± 0.02	11.71 ± 0.39	4.77 ± 0.76	5.65 ± 0.32

Fuc: Fucose; Rham: Rhamnose; Arab: Arabinose; Gal: Galactose; Glu: Glucose; Man: Mannose; Xyl: Xylose; Mannu: Mannuronic acid; Gulur: Guluronic acid; Glucu: Glucuronic acid; SA: Sodium alginate; EF: Ethanol fraction; SF: Solid fraction; LF: Liquid fraction; ND: Not detected.

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
