# Peer review of "Alginate-Derived Elicitors Enhance β-Glucan Content and Antioxidant Activities in Culinary and Medicinal Mushroom, Sparassis latifolia"

_jof, 2020, doi:10.3390/jof6020092_

Round 1

Reviewer 1 Report

I reported all my suggestions and comments in the file attached.

Author Response

Response to Reviewer 1 Comments

I reported all my suggestions and comments in the file attached.

Thanks for all the detailed comments, the following is the changes revised based on the suggestions.

Point 1. Abstract. If you decide to use the term "seaweed" please use it consistently all throughout the text. If you decide to use "brown macroalgae" do the same. Variatio (i.e. different phrases for the same concept/object) is smart, but it may generate confusion in scientific articles.

Response 1: The term "seaweed" has been replaced by "brown algae" throughout the text.

Point 2. Introduction. Please add "but not exclusively"; some species (e.g. S. brevipes) are also related to oak and beech.

Response 2: The phrase "but not exclusively" was added behind "primarily".

Point 3. Introduction. 1) This sentence is unclear: what do you mean?   There isn't only one species in Europe and North America as well as in East Asia. References [3,4] are good: check them and then cross-check updated taxonomy with Mycobank and/or Index Fungorum.

2) Please add species author(s); check them on Mycobank and/or Index Fungorum; add them to every species when first mentioned.

Response 3: The sentence has been rewritten to make it clear: "The Asian Sparassis isolate was originally known as S. crispa, until morphological and molecular studies redefined as S. latifolia [3,4]."

Point 4. Introduction. Sparassis latifolia and S. crispa are well-distinct species: check taxonomy!

Response 4: Sparassis latifolia and S. crispa have been distinguished in the text.

Point 5. Introduction. Be careful: as demonstrated by Gil-Ramirez et al. (2016), fungi can't produce flavonoids; similar phenolics are misidentified by improper analitycal techniques. See below for the reference.

Gil-Ramírez, A., Pavo-Caballero, C., Baeza, E., Baenas, N., Garcia-Viguera, C., Marín, F. R., & Soler-Rivas, C. (2016). Mushrooms do not contain flavonoids. Journal of Functional Foods, 25, 1-13.

Response 5: Thanks for the kind remind, the phrases have been revised: "polysaccharides, terpenoids, phenolic compounds, and glycoproteins"

Point 6. Introduction. Please add more references (1-2).

Response 6: The whole paragraph has been revised and more references were added.

Point 7. Introduction. Be careful when using the phrase "eco-friendly": you say you have to provide alginate first. According to your method, you need formaldehyde and HCl to get it. That's interesting, but not eco-friendly.  You should highlight that this is eco-friendly on condition that alginate achievement is eco-friendly too (alginate industry also uses other processes).

Response 7: The phrase "eco-friendly" was deleted as suggested.

Point 8. Introduction. Why did you add species authors only to some plant species and not to other ones? Please add authors to all plant species or don't add to anyone. Check them on IPNI.

Response 8: The species authors were deleted in all the plant species.

Point 9. Introduction. Please use scientific (latin) names; you can refer to genus or superior categories if you don't point out only one species. Add authors.

Response 9: The scientific names were used: "Ecklonia spp., Laminaria spp., or Sargassum spp."

Point 10.          Material & Methods. Add authors to all mentioned Sargassum species

Response 10: The species authors were added to all Sargassum species.

Point 11.          Material & Methods. Always add a space between the number and "°C". Do it consistently throughout the text.

Response 11: A space was added between the number and "°C" throughout the text (totally 19).

Point 12.          Material & Methods. Were

Response 12: The word "was" changed to "were" throughout the text.

Point 13.          Material & Methods. Please add a reference for your protocol.

Response 13: The reference [32] was added for the protocol of "Molecular Weight Analysis": Alban, S.; Schauerte, A.; Franz, G. Anticoagulant sulfated polysaccharides. I. Synthesis and structure-activity relationships of new pullulan sulfates. Carbohyd. Polym. 2001, 47, 267-276.

Point 14.          Material & Methods. cm-1?

Response 14: "/cm" was changed to "cm-1".

Point 15.          Material & Methods. Please use the scientific (latin) name.

Response 15: The name "Cauliflower Mushroom" was changed to "Sparassis latifolia".

Point 16.          Material & Methods. Reference n°30 does not match. I miss it. Maybe you meant (modified):

Park, H. G., Shim, Y. Y., Choi, S. O., & Park, W. M. (2009). New method development for nanoparticle extraction of water-soluble β-(1→ 3)-D-glucan from edible mushrooms, Sparassis crispa and Phellinus linteus. Journal of Agricultural and Food Chemistry, 57(6), 2147-2154.

Response 16: The reference [23] was corrected: Park, H.; Ka, K.H.; Ryu, S.R. Enhancement of β-glucan content in the cultivation of cauliflower mushroom (Sparassis latifolia) by elicitation. Mycobiology 2014, 42, 41-45.

Point 17.          Material & Methods. Megazyme kit is one of the most widely used tools to assay glucan content - and a relatively easy solution. Anyway, in my experience it can lead to significant error due to the strong hydrolyzation treatment. Take this into account when discussing your data.

Response 17: The β-glucan assay kit (Cat. No. K-YBGL, Megazyme) was chosen and discussed with consideration of its measurement error.

Point 18.          Material & Methods. Did you exactly follow a protocol provided by Sigma? Did you follow another protocol? Please provide a reference for your protocol. Moreover, was this wavelength suggested by your protocol or did you test it by scanning the whole spectrum (if possible by your microplate reader)?

Response 18: The SOD kit (Cat. No. 19160, Sigma-Aldrich) was used and followed the protocol, the wavelength λ = 450 nm was suggested as the protocol, and the reference was added [38]: Odeyemi, S.; Dewar, J. Repression of acetaminophen-induced hepatotoxicity in HepG2 cells by polyphenolic compounds from Lauridia tetragona (L.f.) R.H. Archer. Molecules 2019, 24, 2118.

Point 19.          Material & Methods. Be careful: do you mean standard deaviation or standard error? Please use standard error and correct all your data consistently.

Response 19: Standard error was used as suggested and all data were corrected correspondingly.

Point 20.          Results. This is a latin word: the correct plural is "rationes".

Response 20: The word "ratios" was changed to "rationes".

Point 21.          Results. Why do you discuss this here? Are you going to make a comparison with other species? Please move your comparisons in Discussion and provide a reference. Moreover, check the spell and taxonomy: the currently accepted name for Sargassum vulgare is S. tenuissimum.

Response 21: The name "S. vulgare" was mistaken, it should be "S. horneri".

Point 22.          Results. As a whole as it concerns your FTIR analysis, READ CAREFULLY:

- you can't be so confident when attributing peaks to certain biomolecules, since biomolecules are very similar each other and have complex structure; you can only speculate based on their chemical groups and bonds. FTIR is not so specific in discriminating biomolecules. You can obtain peculiar spectra, but you are not sure they are fingerprint spetra for an algal species. Anyway, this is not the right technique to discriminate so similar molecules within a complex matrix as "solid fraction", "ethanol fraction", ecc.: too many variables are overlapped.

- If you suppose that some peaks are specific features of certain molecules, please provide reference from literature and also provide a more focused figure of fingerprint region.

-Please note I am not going to reject the presence of FTIR analysis in your paper: just revise your data exposition and discussion.

Response 22: Thanks for the suggestion and the phrases have been corrected.

Point 23.          Results. Please use cm-1.

Response 23: "/cm" was changed to "cm-1".

Point 24.          Results. Please use chemical formula consistently thorughout the text.

Response 24: Chemical formula "H2SO4" was used instead of "sulfuric acid".

Point 25.          Results. Please, check the spell of species names! Use AlgaeBase as a support also for taxonomy.

Response 25: All the species names have been checked for the spelling.

Point 26.          Results. The syntax of this sentence is unclear.

Response 26: The sentence was rewritten: The main monosaccharides of alginate from EF, SF, and LF were mannuronic acid, guluronic acid, and glucuronic acid.

Point 27.          Results. You sometimes use "," to mark thousands and sometimes you don't. Please proceed by the same way consistently throughout the text. Personally, I would erase "," because it is unclear outside anglosaxon public.

Response 27: All the "," in the thousands were erased as suggested.

Point 28.          Results. 1) mg g-1

2) please correct decimals based on the standard deviation; as discussed above, check you are using standard error instead of standard deviation in such a table.

Response 28: "mg/g" was changed to "mg g-1". Standard error was used as suggested.

Point 29.          Results. mg ml-1

Response 29: "mg/mL" was changed to "mg mL-1".

  1. Discussion. Pleonastic: cultivation is always human work.

Response 30: The word "artificial" was deleted.

Point 31.          Discussion. Please refer to the original article instead of Wasser et al. (2011) or cite two both.

Response 31: The reference of Wasser et al. (2011) was deleted.

Point 32.          Discussion. READ CAREFULLY:

From this paragraph forward your discussion becomes confused and it looks like a review of findings about the elicitor effects of carbohydrates from algae. Please sum up and, above all, focus on your own work.

Response 32: Paragraph 2-4 in Discussion part were rewritten as suggested, more discussions of our data were included and Table 4 was deleted.

Point 33.          Discussion. You are also referring to fungi, not only to plants; just add a word or phrase to specify.

Response 33: "Plants" were changed to "plants and fungi" as suggested.

Point 34.          Conclusions. This is the valuable result of your study, but remember and please specify here that possible scale-up must take into account an environmentally friendly process to get alginate.

Response 34: Thanks for the kind remind, an environmentally friendly process will be included in future.

Reviewer 2 Report

Thank you for your interesting research and your exhaustive work. There are some important points that need to be carefully revised:

  1. Abstract. The abstract is well-structured and it is self-explanatory. However, is too long. The word counter registers 346 words and the maximum is 200 words for JoF. Please, try to reduce the length of your abstract.
  2. Introduction. Page 2, second paragraph. You described the “β-glucan” as only one β-glucan existing in Sparassis species. However, the existence of only one structure is unlikely to occur. I highly recommend to enrich the introduction part describing the existence of different structures of β-glucans in edible mushrooms, particulary linear 1-3, linear 1-6 and branched 1-3 1,6. Please, utilize and add these useful references: https://doi.org/10.1016/j.carbpol.2019.04.051 ; https://doi.org/10.1016/j.jff.2019.103446 ; https://doi.org/10.1016/j.carbpol.2019.115521
  3. Introduction. Related with the previous comment, after a brief description of the existence of different structures in edible mushrooms, a detailed explanation of the different beta-glucan structures observed in Sparassis spp. (with proper citations) would reinforce the Introduction section.
  4. Material & Methods. In 2.7. Assay of Glucan Content. Are you sure that Megazyme kit allow you to quantify alpha-glucans? Please, check it. As far as I know, the beta-glucan content is an indirect measure calculated substracting [alpha-glucan + glucooligosaccharides + free glucose] to [total glucans]. But, how you measure only alpha glucan content without glucooligosaccharides or free glucose?
  5. Results. In the section “3.5. Alginate Effects on Antioxidant Activities”, could you consider the possibility to calculate statistic correlation between TPC and antioxidant activity?
  6. Discussion. What do you think about the effect of alginate elicitation to beta-glucan structure? I understand that the study is focused on beta-glucan content, but do you think that alginate-derived products elicitation could affect to chain length, molecular weight, degree of branching, etc.?

Author Response

Response to Reviewer 2 Comments

Thank you for your interesting research and your exhaustive work. There are some important points that need to be carefully revised:

Point 1: Abstract. The abstract is well-structured and it is self-explanatory. However, is too long. The word counter registers 346 words and the maximum is 200 words for JoF. Please, try to reduce the length of your abstract.

Response 1: The Abstract was rewritten and shortened to 200 words as suggested.

Point 2: Introduction. Page 2, second paragraph. You described the “β-glucan” as only one β-glucan existing in Sparassis species. However, the existence of only one structure is unlikely to occur. I highly recommend to enrich the introduction part describing the existence of different structures of β-glucans in edible mushrooms, particulary linear 1-3, linear 1-6 and branched 1-3 1,6. Please, utilize and add these useful references: https://doi.org/10.1016/j.carbpol.2019.04.051 ; https://doi.org/10.1016/j.jff.2019.103446 ; https://doi.org/10.1016/j.carbpol.2019.115521.

Response 2: Paragraph 2 was rewritten as suggested and the structures of different β-glucan from mushrooms were supplemented. The references were included as recommend.

Point 3: Introduction. Related with the previous comment, after a brief description of the existence of different structures in edible mushrooms, a detailed explanation of the different beta-glucan structures observed in Sparassis spp. (with proper citations) would reinforce the Introduction section.

Response 3: The structure of Sparassis β-glucan was discussed in detail and proper citations were added as suggested.

Point 4: Material & Methods. In 2.7. Assay of Glucan Content. Are you sure that Megazyme kit allow you to quantify alpha-glucans? Please, check it. As far as I know, the beta-glucan content is an indirect measure calculated substracting [alpha-glucan + glucooligosaccharides + free glucose] to [total glucans]. But, how you measure only alpha glucan content without glucooligosaccharides or free glucose?

Response 4: The β-glucan assay kit (Cat. No. K-YBGL, Megazyme) was chosen for the quantification of total and β-glucan, α-glucan was measured according to McCleary and Draga procedure. The contents of α-glucan were determined using the reagent blank of free glucose.

Point 5: Results. In the section “3.5. Alginate Effects on Antioxidant Activities”, could you consider the possibility to calculate statistic correlation between TPC and antioxidant activity?

Response 5: Thanks for the kind suggestion of the statistic correlation between TPC and antioxidant activity, however due to the different measurement and units of these two activities, it’s difficult to establish direct correlation between them.

Point 6: Discussion. What do you think about the effect of alginate elicitation to beta-glucan structure? I understand that the study is focused on beta-glucan content, but do you think that alginate-derived products elicitation could affect to chain length, molecular weight, degree of branching, etc.?

Response 6: Thanks for the valuable suggestion and we’ll take it into consideration of how alginate elicitation affecting the structure of β-glucan in the Discussion part.